# Hepatitis B infection: Evaluation of demographics and treatment of chronic hepatitis B infection in Northern-western Tanzania

**Mathias Mlewa**[1,2☯*], **Helmut A. Nyawale**[2‡], **Shimba Henerico**[3☯], **Ivon Mangowi**[3‡], **Aminiel Robert Shangali**[1‡], **Anselmo Mathias Manisha**[4‡], **Felix Kisanga**[5‡], **Benson R. Kidenya**[6☯], **Hyasinta Jaka**[7,8‡], **Semvua B. Kilonzo**[8‡], **Mariam M. Mirambo**[2☯], **Stephen E. Mshana**[2☯]

**1** Department of Microbiology and Immunology, Mwanza University, Mwanza, Tanzania, **2** Department of Microbiology and Immunology, Catholic University of Health, and Allied Sciences, Mwanza, Tanzania, **3** Department of Central Pathology Laboratory, Molecular Biology Laboratory, Bugando Medical Centre, Mwanza, Tanzania, **4** Department of Biochemistry and Molecular biology, Mwanza University, Mwanza, Tanzania, **5** Department of Public Health, Mwanza University, Mwanza, Tanzania, **6** Department of Biochemistry and Molecular Biology, Catholic University of Health, and Allied Sciences, Mwanza, Tanzania, **7** Department of Gastroenterology, Bugando Medical Centre, Mwanza, Tanzania, **8** Department of Internal Medicine, Catholic University of Health, and Allied Sciences, Mwanza, Tanzania

☯ These authors contributed equally to this work.
‡ HAN, IM, ARS, AMM, FK, HJ and SBK also contributed equally to this work.
* mathiasmlewa@bugando.ac.tz, mathiasxnl2021@gmail.com

**Data Availability Statement:** All relevant data are within the manuscript.

## Abstract

### Background

Chronic hepatitis B virus (HBV) infection is still a major public health problem. In response to the World Health Organization (WHO), Tanzania implemented immunization and treatment to achieve the eradication of HBV infection by 2030. To achieve this goal, frequent updates of demographic data, antiviral therapy eligibility, and uptake are essential. We therefore evaluated demographic data, antiviral therapy eligibility, and uptake among chronically HBV-infected patients attending at Bugando Medical Centre (BMC), Tanzania.

### Methods

A cross-sectional study enrolled 196 chronic HBV patients from April 23, 2023, to October 10, 2023, at BMC, where 100 and 96 patients were retrospectively and prospectively enrolled, respectively. Study's ethical clearance and permission were observed by the Catholic University of Health and Allied Sciences/Bugando Medical Centre research ethics and review committee and the Bugando Medical Centre management respectively. For all patients, socio-demographic data and whole blood samples were obtained. Full blood picture, alanine and aspartate amino transferases, and HBV viral load parameters were determined. Aspartate-Platelet Ratio Index (APRI) and Fibrosis Four (FIB-4) scores were calculated according to their respective formulas. Therapy eligibility and uptake were evaluated according to the 2015 WHO HBV prevention, treatment, and care guidelines. The data were summarized and analysed using STATA version 15.

**Funding:** The author(s) received no specific funding for this work.

## Results

The median age for all patients was 39 [IQR: 32–47.5] years. Nearly all study patients, 99% (194/196), were older than 20 years old, with significant male dominance (73.5% [144/196] versus 26.5% [52/196]; p<0.0001). Anti-HBV antiviral therapy eligibility was 22.4%, while uptake was 6.8% (3/4), which was significantly lower than the WHO expectation of 80% (p <0.0001).

## Conclusion

Almost all chronically HBV-infected patients attending at BMC were older than 20 years old and were significantly dominated by males. Antiviral therapy uptake was remarkably lower than expected by the WHO towards combating HBV infection by 2030.

## Introduction

Chronic hepatitis infection is still a public concern in Tanzania, with a pooled prevalence of 5.2% (5160) in 100,000 individuals [1]. In Mwanza, North-Western Tanzania, hepatitis B surface antigen (HBsAg) prevalence has been reported to range from 3.0% among pregnant women attending the antenatal clinic (ANC) at Bugando Medical Centre (BMC), Mwanza [2], to 7.0% among health care workers (HCWs) at BMC, Mwanza [3]. Hepatitis B virus (HBV) can cause hepatitis B liver diseases such as liver cirrhosis and hepatocellular carcinoma [4, 5], which have poor treatment outcomes [6]. In 1992, the WHO responded to the threat of HBV infection by summoning every country to integrate HBV vaccination into their universal childhood vaccination programs by 1997 [7]. In response to this summons, Tanzania introduced a free infant and paediatric HBV vaccination in the form of a pentavalent vaccine containing the Diphtheria, Pertussis, Polio, Tetanus, Hepatitis B, and Haemophilus influenzae type b3 (DPT-Heb-Hib3) vaccine in 2003 [8]. HBV vaccination is also available for adults, but not for free, and has mostly been focused on HCWs [9]. In addition to HBV vaccination, the World Health Organization (WHO) introduced the first guidelines for the prevention, treatment, and care of persons living with chronic hepatitis B virus (CHBV) infection in 2015, (Table 1), [10]. Following the release of the WHO guidelines in 2015, several countries have adopted their local HBV prevention, treatment and care guidelines. Among the well-known local guidelines are American Association for the Study of Liver Diseases (AASLD) [11] and European Association for the Study of Liver (EASL) [12]. Tanzania has not yet developed any local guidelines for HBV treatment and therefore it relies on the WHO guidelines for both HBsAg testing and treatment (Fig 1). Although this study was conducted and analysed during the era of 2015 WHO-HBV guidelines, in March 2024, the new 2024 WHO-HBV guidelines were released, which have more eligibility criteria and lower cut-off values [13]. Both of the 2015 and 2024 WHO guidelines criteria are shown in Table 1. In Tanzania, testing for coinfections (HIV, HCV, and HDV) and transient elastography (TE) value estimations are not routinely done. Moreover, some criteria found in the new 2024 guidelines were not assessed at the time of conducting this study. These criteria include family history of liver cancer or cirrhosis, presence of immune suppression, presence of comorbidities and presence of extrahepatic manifestations. Therefore, criteria assessed in this study are based on age, APRI scores, ALT, HBV DNA levels. According to the 2015 WHO guidelines, criteria used were: i) all chronic HBV-infected patients with an aspartate-platelet ratio index (APRI) of ≥2; ii) all chronic

**Table 1. WHO eligibility criteria for starting anti-HBV antiviral therapy.**

| 2015 WHO guidelines | 2024 WHO guidelines |
|---|---|
| (i) Aged > 30 years with APRI score ≤2, persistently abnormal ALT levels and HBV DNA >20, 000 IU/mL | (i) Aged ≥12 years with APRI score of ≥0.5 or TE of >7kPa regardless of HBV-DNA and ALT levels |
| (ii) Adults, adolescents and children with clinical LC or APRI score >2 regardless of ALT levels and HBV DNA levels. | (ii) ≥12 years old with clinical LC or APRI score >1 or TE of >12.5kPa regardless of HBV-DNA and ALT levels |
| (iii) As in (i) above | (iii) Aged ≥12 years a with HBV DNA >2000 IU/mL and ALT level above ULN. Adolescents should have ALT>ULN on at least two occasions in a 6- to 12-month period |
| (iv) HBV/HIV, HBV/HDV and HBV/HCV regardless of the APRI score or HBV DNA or ALT levels. | (iv) HBV/HIV, HBV/HDV and HBV/HCV regardless of the APRI score or HBV DNA or ALT levels. |
| | (v) Family history of liver cancer or cirrhosis regardless of the APRI score or HBV DNA or ALT levels. |
| | (vi) Presence of immune suppression (such as long-term steroids, solid organ or stem cell transplant) regardless of the APRI score or HBV DNA or ALT levels. |
| | (vii) Presence of comorbidities (such as diabetes or metabolic dysfunction––associated steatotic liver disease) regardless of the APRI score or HBV DNA or ALT levels. |
| | (viii) Presence of extrahepatic manifestations (such as glomerulonephritis or vasculitis) regardless of the APRI score or HBV DNA or ALT levels. |
| | (ix) Persistently abnormal ALT levels (defined as two ALT values above the ULN at unspecified intervals during a 6- to 12-month period) |

APRI: Aspartate amino transferase, TE: Transient elastography, kPa: Kilo Paschal, ALT: Alanine amino transferase, HBV DNA: Hepatitis B virus deoxyribonucleic acid, IU: International unit, LC: Liver cirrhosis, ULN: Upper limit normal, HIV: Human immunodeficiency virus, HCV: Hepatitis C virus, HDV: Hepatitis D virus

HBV-infected patients older than 30 years old with a hepatitis B viral load of ≥20,000 IU/mL. Eligible patients are required to take antiviral therapy, which can be either interferon-based or nucleoside/nucleotide-based therapy [10]. Treating eligible patients with anti-retroviral therapy is believed to reduce the risk of developing hepatitis B liver diseases [14, 15]. Also, by suppressing HBV viral load [16], treatment reduces the risk of patients' infectivity [17]. Currently, HBV treatment in Tanzania are available at Bugando Medical Centre since 2009 and Muhimbili National Hospital (MNH) through up-to date ongoing project since 2016 [8]. Although HBV treatment is given for free at MNH to eligible chronically infected patients, patients have to buy their own anti-HBV drugs. This might impose an economical barrier to anti-HBV drug uptake. Moreover, data of chronic hepatitis B infected patients taking anti-HBV drugs are scarce in Tanzania. A recently published study done at Bugando Medical Centre has shown no anti-HBV uptake among the eligible chronic HBV infected patients [18]. Tanzania has no local HBsAg testing, treatment and monitoring algorithm and therefore it relies on the WHO guidelines which is more universal. Tanzania is in line with the WHO-HBV combat efforts through the achievement targets of eliminating HBV infection by 2030 [19]. To achieve this goal, Tanzania needs to be updated frequently on the demographic profile of chronic HBV infected patients as well as anti-HBV antiviral therapy (AVT) eligibility and uptake. It has been almost 20 years since the introduction of the infantile-paediatric HBV vaccination program; however, the distribution of chronic HBV infection among patients younger than and older than 20 years old is not well known in Tanzania. Moreover, the proportion of patients eligible for anti-HBV AVT eligibility and taking anti-HBV AVT according to the 2015 WHO-HBV

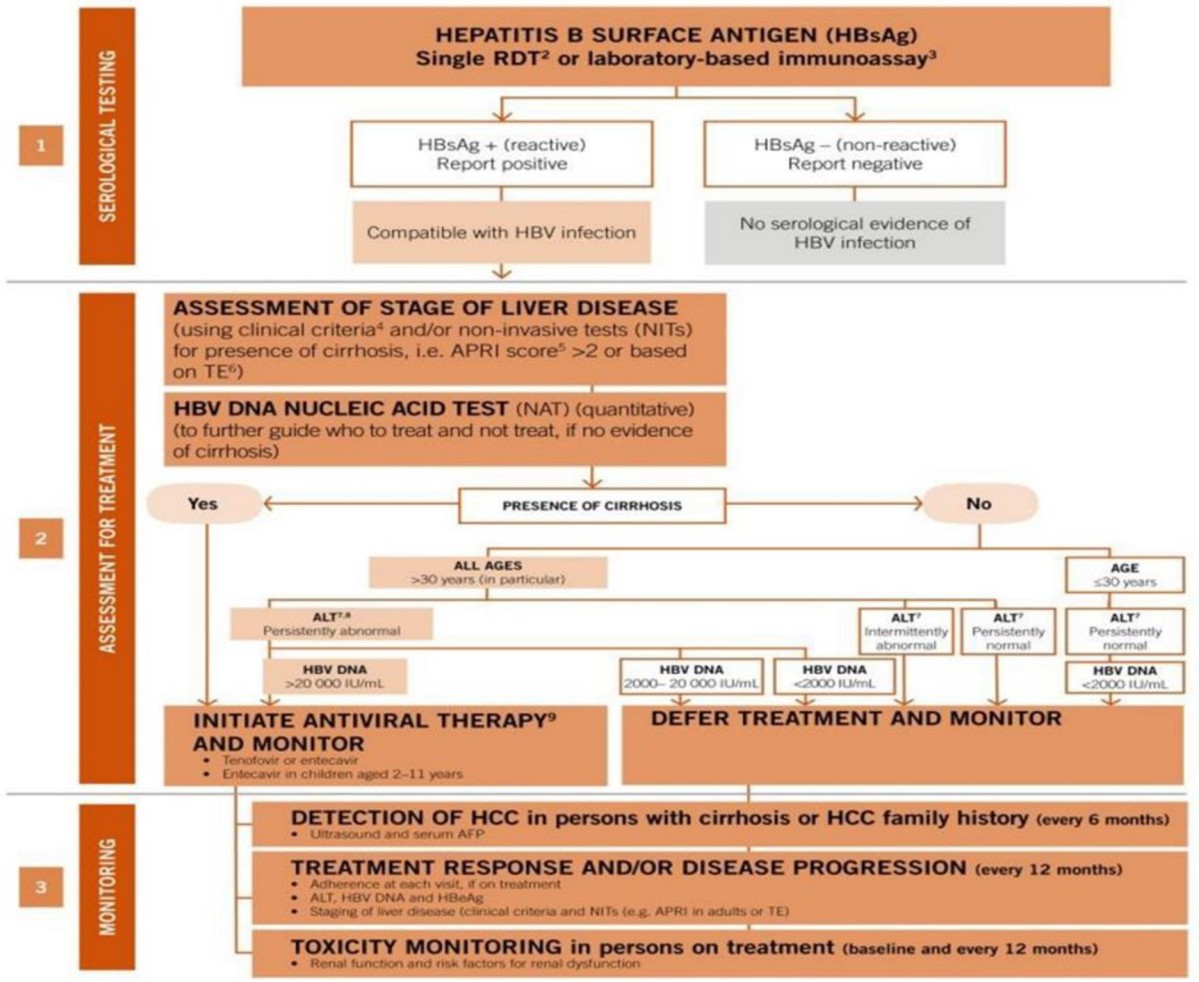

**Fig 1. 2015 WHO guidelines on hepatitis B testing and treatment.**

prevention, treatment, and care guidelines among chronically HBV-infected patients has not been evaluated and known since the introduction of treatment in Tanzania. Therefore, this study provides information on the demographic distribution of chronic HBV infection by age and sex, proportions of anti-HBV AVT eligibility, and uptake according to the 2015 WHO-HBV prevention, treatment, and care guidelines among chronically HBV-infected patients attending the BMC in North-western Tanzania due to the fact that these patients were diagnosed and managed during the era of the 2015 WHO-HBV guidelines.

## Patients, materials and methods

### Ethical considerations

The study was approved by the joint Catholic University of Health and Allied Sciences/ Bugando Medical Centre (CUHAS/BMC) research ethics and review committee **(CREC/663/ 2023)**. In both prospective and retrospective participants, the need for consent was waived by CUHAS/BMC research and review committee. Permission to use the retrospective data from the Bugando Medical records was granted by to conduct the study at Bugando Medical Centre was given by the Bugando Medical Centre management **(AB.286/317/01/PART "M")**. All

retrospective data were accessed with fully anonymity by using their hospital registration number. After retrieving, the participants were given study identification number. The prospective participants were recruited by using study identification number. These study identification numbers were used in all of their hospital tests related to the study.

## Study design, duration and sample size

From 23$^{rd}$ April 2023 to 10$^{th}$ October 2023, we conducted a cross-sectional study at the Bugando Medical Centre, a referral, consultancy, and teaching university hospital. BMC serves a definitive population of about 640 patients with chronic HBV infection annually. The Yamane Taro formula ($n = N/1+N(e)^2$) was used to calculate the minimum sample size. In this formula, n represents the sample size, **N** represents the total population, **e** represents the level of significance and **1** (one) represents a constant value. Yamen Taro formula is used for calculating sample size in a definite study population. Since this study was conducted for a duration of six months, the definitive study population was 320. Therefore, in this study, N was 320 and **e** used was 0.05. On computing these figures on the above formula, a minimum sample size of 178 was obtained. However, in this study 196 patients were enrolled and included in the final analysis. Of these, 100 patients were recruited retrospectively from 23$^{rd}$ April 2023 to 21$^{st}$ May 2023, whereas 96 patients were recruited prospectively from 23$^{rd}$ April 2023 to 7$^{th}$ August 2023.

## Participants

**Prospective patients.** The study enrolled patients with repeatedly confirmed Hepatitis B surface antigen-positive (HBsAg-positive) for more than six months who came for routine full blood picture (FBP), liver enzymes: alanine amino transferase (ALT) and aspartate amino transferase (AST), as well as HBV deoxyribonucleic acid (DNA) levels at Bugando Medical Centre (BMC).

**Prospective sample collection and processing.** All laboratory procedures were performed in the ISO-certified Department of Central Pathology Laboratory of Bugando Medical Centre. Aseptic techniques were used to collect whole blood from the 96 prospective patients. Approximately 15 mL of whole blood from each patient were drawn from each patient for analysis of the full blood picture, liver enzymes alanine and aspartate transferase, as well as the hepatitis B viral load. The full blood picture was analysed with whole blood drawn into ethylenediaminetetraacetic acid tubes using the Sysmex XN-1000 hematology analyser, while the alanine and aspartate amino transferases were analysed with whole blood drawn into plain tubes using COBAS integra 400. Plasma for HBVL was prepared by using whole blood drawn into an ethylenediaminetetraacetic acid tube, followed by centrifugation at 1500 revolutions for 10 minutes, and then about 2000 μL of plasma was drawn into well-labelled cryotubes and stored at -80˚C while waiting for batch hepatitis viral load testing. During batch HBV DNA levels testing, about 1000μL of plasma per patient were used in a real-time PCR assay using the automatic COBAS® AmpliPrep/COBAS® TaqMan® HBV Test, v2.0, Roche Diagnostics, with a quantification linear range of 20 to 1.7 x 10$^8$ IU/mL [20]. PCR amplification of HBV DNA was achieved using the HBV DNA kit with Lot No. J1493600000. Results of HBV DNA batch testing was either target not detected (TND), HBV DNA levels <20 IU/mL, with the linear range of 20 to 1.7 x 10$^8$ IU/mL or > 10$^8$ IU/mL. The prospective participants were tested according to their routine clinic visits request and payments managed by the patient themselves. Although, HDV test is not available in Tanzania, HIV and HCV tests were not routinely requested. Therefore HDV, HIV and HCV were not included in this study. All the HBV DNA levels, FBP, and liver enzymes (ALT and AST) results were directly entered into a Microsoft Excel sheet version 16.

**Retrospective participants.** The study also included all retrospective participants who were confirmed HBsAg-positive patients for more than six months in the BMC HBV medical records with a complete set of demographic and laboratory data, including age, gender, FBP, ALT, AST, and HBV DNA levels. The study excluded HBsAg-positive patients with missing either one of the demographics, FPB, liver enzymes, HBV DNA levels, or all of them.

**Retrospective participants' data extraction and processing.** Data extracted from retrospective participants were demographic data, FBP, Liver enzymes (AST and ALT). Two hematology analysers had been used to analyse FBP; 82 had been performed by Sysmex XN-1000 and 18 by Dymind-DH 76. Among the AST data, 74 of them had been performed by COBAS integra 400 and 26 by Cobas E 601. Eight six (86) ALT data had been performed by Cobas integra 400 and 14 by Cobas E 601. All HBV DNA levels had been analysed by using real-time PCR assay using the automatic COBAS® AmpliPrep/COBAS® TaqMan® HBV Test, v2.0, Roche Diagnostics, with a quantification linear range of 20 to $1.7 \times 10^8$ IU/mL [20]. However, the kit Lot numbers used for real-time PCR amplification were not readily identified. HBV DNA levels recorded were either target not detected (TND), HBV DNA levels <20 IU/mL, with the linear range of 20 to $1.7 \times 10^8$ IU/mL or $> 10^8$ IU/mL. Anti-hepatitis B virus (anti-HBV) antiviral therapy (AVT) uptake data were extracted from the Bugando Medical Centre (BMC) medical record database. Although, HIV and HCV testings are done in Tanzania, most of retrospective participants had not been tested for these parameters. Therefore, these data were no considered for anti-HBV drug eligibility selection. The extracted data were entered directly into an Excel sheet version 16.

**Data management and analysis.** Both prospective and retrospective patients' data (demographics, HBV DNA levels, ALT, and AST laboratory results) were directly entered into a Microsoft Excel sheet version 16. HBV DNA levels of the target not detected; <20IU/mL and $>1.7 \times 10^8$ IU/mL were treated as 0, 20, and $1.7 \times 10^8$ IU/mL respectively, for statistical analysis. Aspartate amino transferase-platelet ratio index (APRI) score [21] and Fibrosis Four score (FIB-4) [22] were calculated according to their respective formula. After cleaning and coding, these data were subsequently transferred to STATA software version 15 for analysis. The percentage was used to summarize categorical variables, which included sex, age groups, and outcome cut off values. The median with Inter Quartile Range [IQR] was used to summarize continuous variables, which included the patients' age, HBV DNA levels, APRI score, and FIB-4 score. The patients were grouped into two age groups. Group I included patients younger than or equal to 20 years old, and Group II included patients older than 20 years old. The age cut off of 20 years was chosen because the free infantile-paediatric hepatitis B (HBV) vaccination began in January 2003; making a lapse of almost 20 years. Hence, all patients in Group I with age of 20 years old and below were presumed to be from the vaccinated cohort. The anti-hepatitis B antiviral therapy eligibility and uptake were also analysed based on the 2015 WHO-HBV prevention, treatment, and care guidelines. We used a two-sample proportion test to analyse the significance of the difference in proportions of chronic hepatitis B infection between Groups I and II, between males and females, and between the AVT uptake of this index study and that recommended by the WHO.

## Results

### Participants' social demographic and clinical data

A total of 196 chronically HBV-infected patients were enrolled in the study. The median age of patients was 39 [IQR: 32–47.5] years old. Nearly all study patients, 99% (194/196), were older than 20 years old (Group I), with a significant male predominance (73.5% [144/196] versus 26.5% [52/196]; p <0.0001). The patients analyzed were from 11 regions of Tanzania, and almost half of the patients (44.4%) were from the Mwanza region (Table 1).

Study participants had HBV DNA levels with a median of 979 [185.5–8457.5] IU/mL, and the majority, 160/196 (81.6%) of the patients, had HBVL below 20,000 IU/mL. The median ALT was 26.0 [18.6–39.4] IU/L, and the majority of them, 76.5% (150/196), had ALT values ≤ 41 IU/L. The median of AST values was 25.8 [IQR: 21.6–39.6] IU/L, and the majority of them, 75.5% (148/196), had AST values ≤40 IU/L. The median platelet counts were 198 [IQR: 154.5–241] cells/μL, and most of them, 79.6% (156/196), had values within the normal range of between 150 cells/μL and 500 cells/μL. These patients had a median APRI score of 0.34 [IQR: 0.25–0.68], while their median FIB-4 score was 1.1 [IQR: 0.8–1.8] (Tables 1 and 2).

**Anti-HBV ARV eligibility and uptake status.** Out of 196 patients, a minority of 22.4% (44/196) were eligible for anti-hepatitis B virus (anti-HBV) antiretroviral therapy (AVT) according to the 2015 World Health Organization (WHO) hepatitis B virus (HBV) prevention, treatment, and care guidelines. Among the eligible patients, 77.3% (34/44) were eligible due to having a HBV DNA levels > 20,000 IU/mL and an age above 30 years, while 22.7% (10/44) were eligible due to having an aspartate amino transferase-platelet ratio index (APRI) ≥2 score. No eligible patient was due to HBV co-infection with either HDV, HIV or HCV as these data were not available or not tested as explained the methodology section. Among the eligible patients, AVT uptake was 6.8% (3/44) which was significantly lower than the WHO expectation of 80% towards elimination of hepatitis B infection by 2030, p<0.0001. However, the eligibility proportion was 30.6% based on the new 2024 guidelines with AVT uptake of 3.3%. The eligible participants with age ≥12 years and APRI score of ≥0.5 regardless of HBV-DNA and ALT levels were 95% (57/60) and with age ≥12 years and HBV DNA >2,000 IU/mL and an ALT level above the upper limit of normal (ULN) were 5% (3/60). All participants aged ≥12 years and with APRI score of >1 regardless of HBV-DNA and ALT levels were also found to be a group of participants with age ≥12 years and APRI score of ≥0.5 regardless of HBV-DNA and ALT levels. Persistently abnormal ALT levels (defined as two ALT values above the ULN at unspecified intervals during a 6- to 12-month period) was not assed due to the nature of the study. Also, the overall antiviral therapy uptake which included eligible and non-eligible antiviral uptake was 6.1% (12/196) and among these, majority, 75% (9/12) uptake was non-eligible (Table 3). Among the eligible patients on antiviral therapy, two patients were on Tenofovir Disoproxil Fumarate, and one patient was on Tenofovir Alafenamide Fumarate. In the non-eligible group of patients who were on antiviral therapy, five of them were on Tenofovir Disoproxil Fumarate; three of them were on a fixed drug combination antiretroviral drug with Emtricitabine and Tenofovir Disoproxil Fumarate (TRUVADA); and one patient was on a fixed drug combination antiretroviral drug with Tenofovir Disoproxil Fumarate, Lamivudine, and Dolutegravir (Table 4). Furthermore, among the eligible patients with high HBV DNA levels above 20,000 IU/mL, the majority of them, 91.12% (31/34) were not on anti-HBV antiretroviral therapy (Table 5).

## Discussion

We believe that this is the first study to report on evaluation of demographic data and treatment of chronic hepatitis B (CHBV) infection since the introduction of the national infantile-paediatric HBV vaccination program and HBV treatment in Tanzania. Our results have shown that, nearly all chronic hepatitis B-infected patients were older than 20 years. This might be due to the effect of the national infantile-pediatric HBV vaccination program that started about 20 years ago. On the other hand, the study showed remarkably lower anti-HBV antiviral therapy uptake than the expectation of the WHO towards the elimination of HBV by 2030 among the eligible patients.

The reduction of proportion of chronic HBV infection in patients older than 20 years old is probably due to the effect of the national infantile-pediatric HBV vaccination, which started in

**Table 2. Demographic and clinical data: Proportion/median and percentage/interquartile range [IQIR].**

| Characteristics (Variables) | Number (n)/ (Percentage (%)) | Median | Interquartile Range [IQR] |
|---|---|---|---|
| **Age** | | 39 | 32–47.5 |
| **Age groups** | | | |
| Age ≤ 20 years old, | 2 (1.0) | | |
| Age > 20 years old | 194 (99.0) | | |
| **Gender** | | | |
| Female | 52 (26.5) | | |
| Female age | | 38.5 | 33–47.5 |
| Male | 144 (73.5) | | |
| Male age | | 39 | 31–47.5 |
| **Residence (Region)** | | | |
| Mwanza | 87 (44.4) | | |
| Mara | 26 (13.3) | | |
| Shinyanga | 23 (11.7) | | |
| Geita | 20 (10.2) | | |
| Other regions | 40 (20.4) | | |
| **Clinical characteristics** | | | |
| **HBV DNA levels (IU/mL)** | | 979 | 185.5–8457.5 |
| **HBV DNA levels groups (IU/mL)** | | | |
| HBV DNA levels ≤20,000 | 160 (81.1) | 81.1 | |
| HBV DNA levels >20,000 | 36 (18.4) | 18.4 | |
| **ALT (IU/L)** | | 26 | 18.6–39.4 |
| **ALT groups (IU/L)** | | | |
| ALT ≤41IU/L | 150 (76.5) | 76.5 | |
| ALT >41 | 46 (23.5) | 23.5 | |
| **AST (IU/L)** | | 25.8 | 21.6–39.6 |
| **AST groups (IU/L)** | | | |
| AST ≤40 | 148 (75.5) | 75.5 | |
| AST >40 | 48 (24.5) | 24.5 | |
| **Platelets (cells/μL)** | | 198 | 154.5–241 |
| **Platelets groups (cells/μL)** | | | |
| Platelets < 150 | 40 (20.4) | 20.4 | |
| Platelets 150–500 | 156 (79.6) | 79.6 | |
| **Fibrosis markers** | | | |
| **APRI score** | | 0.34 | 0.25–0.68 |
| **APRI score groups** | | | |
| APRI ≤2 | 174 (88.8) | 88.8 | |
| APRI >2 | 22 (11.2) | 11.2 | |
| **FIB-4 score** | | 1.1 | 0.8–1.8 |
| **FIB-4 score groups** | | | |
| FIB-4 ≤2 | 154 (78.6) | 78.6 | |
| FIB-4 >2 | 42 (21.4) | 21.4 | |

HBV DNA: Hepatitis deoxyribonucleic acid, ALT: Alanine Amino Transferase, AST: Aspartate Amino Transferase, THR: Thrombocytes, IQR: Interquartile Range, APRI: Aspartate–Platelet Ratio Index, FIB-4: Fibrosis Index Based on 4 criteria.

**Table 3. Eligibility of anti-HBV ARV therapy according to the WHO-2015 guidelines and anti-HBV ARV uptake status.**

| Anti-HBV antiretroviral therapy eligibility status (N = 196) | Number (n) | Percent (%) |
|---|---|---|
| Eligible patients | 44 | 22.4 |
| Non-eligible patients | 152 | 77.6 |
| **Eligibility by criteria (N = 44).** | | |
| High HBV DNA levels ≥20,000 IU/mL and age above 30 years | 34 | 77.3 |
| High aspartate amino transferase-platelet ratio index (APRI) ≥ 2 | 10 | 22.7 |
| **Eligible patients' anti-HBV antiretroviral therapy status (N = 44).** | | |
| On anti-HBV antiretroviral therapy | 3 | 6.8 |
| Not on anti-HBV antiretroviral therapy | 41 | 93.2 |
| **Eligible patients with HBVL ≥20,000IU/ml on anti-HBV antiretroviral therapy (N = 34)** | | |
| On anti-HBV antiretroviral therapy | 3 | 8.8 |
| Not on anti-HBV antiretroviral therapy | 31 | 91.2 |
| **Overall anti-HBV antiretroviral therapy uptake status (N = 196)** | | |
| On anti-HBV antiretroviral therapy | 12 | 6.1 |
| Not on anti-HBV antiretroviral therapy | 184 | 93.9 |
| **Total overall patients on anti-HBV antiretroviral therapy (N = 12)** | | |
| Eligible patients | 3 | 25.0 |

2003 in Tanzania [8]. This reduction in HBV infection in the population with an age equal to an age of the HBV vaccination program has also been reported in Iran [23]. In this this current study, majority of the study participants 73.5% (144/196) were sexually active male with median age of 39 [31–47.5] years old. These findings are comparable with several other global reports on sexual activeness and male dominance in chronic HBV infection [1]. Taken together, these findings of less chronic HBV infection in paediatrics and young adults and more middle-aged, sexually active patients with chronic HBV infection indicate that the main mode of HBV transmission in the country is the horizontal route, particularly through sexual activities, rather than the vertical (intrauterine and perinatal) transmission route, as previously thought in endemic countries, including Tanzania [24]. The sexual transmission route can also be supported by the predominance of genotypes A, D, and E in Tanzania [25], which are more transmitted by the horizontal route than by the vertical route [26]. Moreover, HBV vaccination prevents HBV infection rather than clearing infection. Infantile-pediatric HBV vaccination is given after birth and not intrauterine [8]. Therefore, from this study, we speculate that there could be more horizontal sexual transmission that is contributed more by male

**Table 4. Types anti-HBV antiretroviral drugs used.**

| Overall patients taking antiviral drugs (N = 12) | Drugs used | Number (n) | Percent (%) |
|---|---|---|---|
| In the eligible patients | TDF | 2 | 66.7 |
| | TAF | 1 | 33.3 |
| In the non-eligible patients | TDF | 5 | 55.6 |
| | TRUVADA | 3 | 33.3 |
| | TLD | 1 | 11.1 |

TDF: Tenofovir disoproxil fumarate, TAF: Tenofovir alafenamide fumarate, TLD: Tenofovir-Lamivudine-Dolutegravir, ARV: Antiretroviral, TRUVADA: Fixed antiviral drug combination composed of emtricitabine (FTC) and tenofovir disoproxil fumarate (TDF).

**Table 5. Eligibility of anti-HBV ARV therapy according to the WHO-2024 guidelines and anti-HBV ARV uptake status.**

| Anti-HBV antiretroviral therapy eligibility status (N = 196) | Number (n) | Percent (%) |
|---|---|---|
| Eligible patients | 60 | 30.6 |
| Non-eligible patients | 136 | 69.4 |
| **Eligibility by criteria (N = 60).** | | |
| Participants aged ≥12 years and with APRI score of ≥0.5 regardless of HBV-DNA and ALT levels | 57 | 95 |
| Participants aged ≥12 years and with HBV DNA >2,000 IU/mL and an ALT level above the upper limit of normal (ULN) | 3 | 5 |
| **Eligible patients' anti-HBV antiretroviral therapy status (N = 60).** | | |
| On anti-HBV antiretroviral therapy | 2 | 3.3 |
| Not on anti-HBV antiretroviral therapy | 58 | 96.7 |

individuals in our local setting, which might signify a need to target this sexually active group in preventive measures such as HBV vaccination.

An anti-HBV antiviral therapy eligibility of 22.5% and 30.6% according to 2015 and 2024 WHO guidelines were observed in this study. This eligibility proportion was higher than the 4·4% and 9.7% reported in the Gambian study, which was conducted among community and blood donors using the 2015 WHO guideline, respectively. Community and blood donors have more individuals in the early stages of HBV infection and hence are unlikely to be eligible for antiviral therapy (AVT) compared to the clinical population. The eligibility proportion in this study was also higher than the 10.2% which was previously reported in a largely community (62.9%) composed study population in Zambia conducted according to the 2015 WHO guidelines [27]. Another study using the TREAT-B algorithm guidelines reported an eligibility rate of 56.8% [28], which is higher than that observed in the present study; the difference lies in the criteria used. TREAT-B algorithm guidelines have high sensitivity for identifying patients eligible for anti-HBV antiretroviral therapy [29] compared to the 2015 WHO-HBV guidelines. Moreover, the TREAT-B algorithm guidelines are designed to overcome the limited access to HBV infection diagnostic and evaluation tools in Africa [30]. However, the eligibility proportion of this index study was in accordance with other previous study report (23%) conducted according to the 2015 WHO guidelines [31]. Eligibility proportion of this index study by using the 2024 WHO guidelines [13] was 30.6% which reflects the increase in number of eligibility criteria compared to the old 2015 WHO guidelines.

In our study, anti-HBV antiretroviral therapy uptake observed in this study was 6.8% as per the 2015 WHO-HBV prevention, treatment, and care guidelines. This anti-HBV antiretroviral therapy uptake in the present study was far below the WHO-HBV achievement target of treating 80% of all anti-HBV antiretroviral therapy-eligible patients [19]. The anti-HBV antiviral therapy uptake observed in our study was also far lower than 62.8%, which was reported in a previous study in London [31] which is one of the first world countries. In contrast Tanzanians, citizens in London are likely to be economically capable and feasible of accessing these anti-HBV antiviral drugs. A previous study in Botswana involving HIV/HBV coinfection reported a higher uptake of AVT of 84%, however, the treatment uptake was primarily due to HIV infection [32]. The overall percentage of patients with anti-HBV antiretroviral therapy uptake in the whole population of 196 patients was 6.1%, with an eligible uptake rate of 25%. With respect to non-eligible patients, most of them used different antiretroviral drugs, probably indicating a different and non-formal source of anti-HBV antiretroviral drugs, which was not easily identified in our study population. This low anti-HBV antiretroviral therapy uptake

could be due to the antiretroviral drug cost barrier. Most drugs used for HBV infection treatment are the same as those used for HIV infection treatment. However, in the case of HIV infection, these drugs are offered for free but have to be bought for HBV infection treatment, except at MNH [8]. Anti-HBV antiretroviral drug costs have been a major barrier to drug uptake in both low- and high-income countries [33]. WHO has recommended strategies to encourage countries to develop actions that can control HBV infection and diseases, including ensuring the availability and accessibility of anti-HBV drugs [34]. China in its pilot study, it reduced the cost barrier and resulted into increase of uptake of TDF and ETV [35]. Although Tanzania is among the low- and middle-income countries, with many of its citizens having low to moderate economic capacity [36], it should join the global strategies of controlling hepatitis B infection and disease by ensuring ease availability and accessibility of anti-HBV drugs [34]. The anti-HBV antiviral uptake was much lower (3.3%) than that obtained from by using the old 2015 WHO guideline. This might be to increase in eligibility criteria while uptake remains constant in this index study.

Additionally, the present study has observed that 18.4% of patients had a high hepatitis B virus DNA level of $\geq$20,000 IU/mL. The proportion in our index study was higher than 9.8%, from a study conducted in HBV/HDV coinfected population in Mauritania [37]. Contrast to this current study, HBV/HDV is associated with low HBV replication [38]. A previous study conducted in HIV/HBV co-infected patients who were starting anti-HIV antiretroviral therapy involving Mozambique and Zambia reported a higher proportion of 49.4% [39] than in this current study. HIV has been reported to suppress the immune system, allowing for high replication of HBV [40]. This justifies the high proportion of patient with HBV DNA levels os >20,000IU/mL observed in that previous study involving Mozambique and Zambia. In the present study, the HIV statuses of the patients were not known. Contrast to this current study, a France national surveillance study with most participants likely to be in their early stage of HBV infection with high viral replication reported a high proportion of patients with HBV DNA levels $\geq$20,000 IU/mL [41]. Among the patients with high HBVL >20,000 IU/mL in our study, only 8.8% (3/34) were on anti-HBV antiviral drugs. It has been hypothesized that high HBV DNA levels above 20,000 IU/mL could be associated with a high risk of developing HBV liver-related diseases such as liver cirrhosis and hepatocellular carcinoma [42, 43]. Therefore, the majority of the eligible patients with high HBV DNA levels above 20,000 IU/mL in this current study, (91.2%) are at high risk of developing hepatitis B liver-related diseases such as liver cirrhosis and hepatocellular carcinoma as they are not taking anti-HBV antiretroviral therapy.

Moreover, this current study has observed a significant number of patients, 20.4%, with platelets below 150 cells/μL. This was lower than what has been reported previously, 6.5% in Taiwan [44]. The difference could be due to the platelet cut off value used: a platelet cut off value of $\leq$100 or $\leq$50 cells/μL versus $\leq$150 cells/μL in the Taiwan study versus this index study, respectively. It was almost similar to a previous study done in Iran, which reported a proportion of 28.3% [45]. The similarity could be due to similar platelet cut off values used of $\leq$150 cells/μL.

In this current study, some limitations were observed. First, the classification of chronic hepatitis B infection was based on records of repeated HBsAg positive results for more than six months in the BMC HBV infection database rather than the presence of HBsAg-IgG antibodies and the absence of HBsAg-IgM. HBsAg-IgM and HBsAg-IgG are not routinely done at Bugando Medical Centre (BMC). This could have excluded many potential patients, whom might have resulted in different results in the study analysis. Secondly, HIV/HDV/HCV infection testing and transient elastography (TE) value estimation in HBV infected patients are not routinely done at BMC. This might have resulted into missing of more eligible patients.

Thirdly, the sample size used was calculated based on six months of the study duration. This might be lower than what is required to characterize the definitive population served annually at BMC. This might have given lower or higher estimates of anti-HBV therapy eligibility and uptake. Lastly, at the time of designing and planning this current study, the new 2024 WHO guidelines were not yet released. This made this study to be more centred on the 2015 WHO guidelines.

In conclusion, almost all chronically hepatitis B-infected patients attending at BMC were older than 20 years old and were significantly dominated by males. The reduction of proportion of chronic HBV infection in participants younger than 20 years old might be due to the effect of infantile-paediatric HBV vaccination program. This could be indicating that the Ministry of Health of Tanzania should sustain the infantile-paediatric HBV vaccination program and expand it to the non-paediatric population. Antiviral therapy uptake was remarkably lower than what is expected by the WHO to combat HBV by 2030. This alarms the country to devise and implement strategies that can increase the availability, ease of accessibility, and usage of anti-HBV antiviral drugs among chronically HBV-infected patients. We also call upon the follow-up study on anti-HBV antiviral therapy eligibility and uptake that addresses the limitations observed in this study but also follows the new 2024 WHO guidelines.

## Acknowledgments

The authors would like to acknowledge the funding support provided by Mwanza University for this work as part of the master thesis. The authors would also like to acknowledge Mr. Teonas A. Nyalu and Barnaba Sospeter from the department of medical records at Bugando Medical Centre for their support during data extraction.

## Author Contributions

**Conceptualization:** Mathias Mlewa, Helmut A. Nyawale.

**Data curation:** Mathias Mlewa, Helmut A. Nyawale, Ivon Mangowi, Aminiel Robert Shangali, Anselmo Mathias Manisha, Benson R. Kidenya, Hyasinta Jaka, Semvua B. Kilonzo.

**Formal analysis:** Mathias Mlewa, Helmut A. Nyawale, Shimba Henerico, Ivon Mangowi, Aminiel Robert Shangali, Anselmo Mathias Manisha, Felix Kisanga.

**Investigation:** Mathias Mlewa, Shimba Henerico.

**Methodology:** Mathias Mlewa, Shimba Henerico, Benson R. Kidenya, Hyasinta Jaka, Semvua B. Kilonzo.

**Supervision:** Mariam M. Mirambo, Stephen E. Mshana.

**Writing – original draft:** Mathias Mlewa, Helmut A. Nyawale, Shimba Henerico.

**Writing – review & editing:** Mathias Mlewa, Aminiel Robert Shangali, Anselmo Mathias Manisha, Felix Kisanga, Benson R. Kidenya, Hyasinta Jaka, Semvua B. Kilonzo, Mariam M. Mirambo, Stephen E. Mshana.

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
