## [Decision Letter · Decision Letter 0]

23 Jun 2024

PONE-D-24-17908Hepatitis B infection in Lake Zone Tanzania: Demographics, antiretroviral therapy eligibility and antiviral therapy uptake among chronic hepatitis B patients at a tertiary hospital in North-Western TanzaniaPLOS ONE

Dear Dr. Mlewa,

Thank you for submitting your manuscript to PLOS ONE. After careful consideration, we feel that it has merit but does not fully meet PLOS ONE’s publication criteria as it currently stands. Therefore, we invite you to submit a revised version of the manuscript that addresses the points raised during the review process.

We look forward to receiving your revised manuscript.

Kind regards,

Jason T. Blackard, PhD

Academic Editor

PLOS ONE

Journal Requirements:

Additional Editor Comments:

This is a cross-sectional study of HBV prevalence and treatment eligibility in Tanzania.  Given the burden of HBV in many resource-limited settings, foundational studies such as this one are needed.  However, the written requires significant improvement to make the manuscript methods and results clear and concise.  Editing by a native English speaker and/or professional editing service is needed.

Many complicated sentence structures should be re-written for clarity.  For example,

·      Among the efforts to prevent HBV infection and therefore reduce the risk of developing chronic HBV infection, in 1992, the WHO summoned every country to integrate HBV vaccination into their universal childhood vaccination programs by 1997 [7] so as to reduce the risks of acquiring HBV infection and therefore acute hepatitis B which can lead to chronic hepatitis B infection. 

·      Our results have shown that, although nearly all chronic hepatitis B-infected patients were older than 20 years, which was probably due to the effect of the national infantile-pediatric HBV vaccination program that started about 20 years ago, anti-HBV antiviral therapy uptake among the eligible patients according to the 2015 WHO-HBV prevention, management, and care guidelines for HBV infection was remarkably lower than the expectation of the WHO towards the elimination of HBV by 2030. 

This statement implies that HBV treatment rollout was not country-wide. Is that indeed the case?

·      Tanzania introduced HBV treatment in 2009 at the Bugando Medical Centre and in 2016 at Muhimbili National Hospital [8]. 

Batch testing for HBV DNA is mentioned in the Methods.  However, whether positive pools were then de-complexed and tested individually is not stated.  This is essential for accurate determination of HBV DNA prevalence and levels.

Delete this sentence:

·      Furthermore, the patients were categorized into male and female groups based on sex. 

**Sex** should be replaced with gender throughout the manuscript and tables.

**HBVL** should be replaced with HBV DNA.

The HBV DNA cutoff of **20000** should be replaced with 20,000 throughout the manuscript and tables.

Study limitations – there are several – should be clearly stated in the Discussion.  The study population is modest in size.

The Discussion is quite long and should be shortened / streamlined.

Reviewers' comments:

Reviewer's Responses to Questions

**Comments to the Author**

1. Is the manuscript technically sound, and do the data support the conclusions?

Reviewer #1: No

Reviewer #2: Yes

2. Has the statistical analysis been performed appropriately and rigorously? 

Reviewer #1: No

Reviewer #2: Yes

3. Have the authors made all data underlying the findings in their manuscript fully available?

Reviewer #1: Yes

Reviewer #2: Yes

4. Is the manuscript presented in an intelligible fashion and written in standard English?

Reviewer #1: Yes

Reviewer #2: Yes

5. Review Comments to the Author

Reviewer #1: Title: very long. can be shortened and made more explicit

Methods: The methods section combines a chart review with newly recruited patients. This creates a problem as the methods section technically must be reproducible. There is no way of knowing if the same laboratory methods, under similar conditions was used to generate data collected from the chart review. The study can be salvaged by describing either of the groups independently as two studies and not combining their data.

Data Analysis, Discussion and Conclusion: Due to the anomaly in the methods section, the data analysis , discussion and conclusion are not valid.

Reviewer #2: The study provides important results for a critical area in the African region. Below are some comments and questions for a better understanding of the study.

Introduction

1. The general situation of HBV treatment in Tanzania needs to be clarified. Are Bugando Medical Center and Muhimbili National Hospital the only sites that treat HBV in Tanzania?

2. Is there any information on the number of individuals on treatment among those diagnosed with HBV in Tanzania?

Methodology

3. I suggest that you briefly explain the principle of the formula used to calculate the sample size, including the numbers.

4. Is there information in the database about testing for other pathogens such as HIV, HCV, and HDV? Is testing routine in Tanzania?

5. Is there data on the algorithm used for HBsAg testing in the patients included in the study? Which test is used?

Results

6. Tables 1 and 2 could be merged into a single table. This would give a better view of the results.

7. Tables 1 and 2 can have two columns. One with the characteristic and the other with the absolute number and percentage, just indicate it in the column legend.

8. Other factors such as co-infection with HBV, HDV, and HCV were not taken into account in the eligibility for treatment. If there is no data, it is important to discuss it.

Discussion

9. The WHO recommendations for treatment were updated in 2024. The study was carried out under the 2015 recommendations, but it is also important that the results are discussed in the context of the new recommendations.

6. PLOS authors have the option to publish the peer review history of their article (what does this mean?). If published, this will include your full peer review and any attached files.

Reviewer #1: No

Reviewer #2: No

---

## [Author Response · Author response to Decision Letter 0]

4 Jul 2024

Reviewers’ responses

Requirements/comments Response to the requirement/comment Location

Journal Requirements 

1. Manuscript meets PLOS ONE's style requirements, including those for file naming Responded accordingly Elsewhere in the manuscript

2. Concerning participants’ consent Responded accordingly Page 2,9

3. Delete this sentence: Furthermore, the patients were categorized into male and female groups based on sex Sentence deleted accordingly Page 13

4. Sex should be replaced with gender throughout the manuscript and tables Responded accordingly Page 15 and elsewhere in the manuscript

5. HBVL should be replaced with HBV DNA.

 Responded accordingly Elsewhere in the manuscript

6. The HBV DNA cutoff of 20000 should be replaced with 20,000 throughout the manuscript and tables.

 Responded accordingly Elsewhere in the manuscript

7. Many complicated sentence structures should be re-written for clarity. For example,

7a. Among the efforts to prevent HBV infection and therefore reduce the risk of developing chronic HBV infection, in 1992, the WHO summoned every country to integrate HBV vaccination into their universal childhood vaccination programs by 1997 [7] so as to reduce the risks of acquiring HBV infection and therefore acute hepatitis B which can lead to chronic hepatitis B infection. Responded accordingly Page 3

7b. Our results have shown that, although nearly all chronic hepatitis B-infected patients were older than 20 years, which was probably due to the effect of the national infantile-pediatric HBV vaccination program that started about 20 years ago, anti-HBV antiviral therapy uptake among the eligible patients according to the 2015 WHO-HBV prevention, management, and care guidelines for HBV infection was remarkably lower than the expectation of the WHO towards the elimination of HBV by 2030. 

 Rephrased accordingly Page 20

This statement implies that HBV treatment rollout was not country-wide. Is that indeed the case?

· Tanzania introduced HBV treatment in 2009 at the Bugando Medical Centre and in 2016 at Muhimbili National Hospital [8]. 

 YES. Readdressed more informatively. Page 4, 5

Batch testing for HBV DNA is mentioned in the Methods. However, whether positive pools were then de-complexed and tested individually is not stated. This is essential for accurate determination of HBV DNA prevalence and levels.

 The test done is quantitative real-time PCR and not qualitative. All participants were known HBV positive patients attending at Bugando Medical Centre-HBV clinic. The results were either target not detected (TND) which would be treated as negative in qualitative PCR. TND was treated as zero value. Other results were <20IU/mL, 20IU/mL to 1.7 X108IU/mL or >1.7 X108IU/mL.

It well explained accordingly Page 11

Study limitations – there are several – should be clearly stated in the Discussion. The study population is modest in size.

 Study limitations revised and discussed accordingly Page 26, 27

The Discussion is quite long and should be shortened / streamlined.

 The discus section sectioned revised and shortened Page 20-27

Review Comments to the Author 

Reviewer #1: 

1. Title: very long. can be shortened and made more explicit

 Title shortened and improved accordingly Page 1

2. Methods: The methods section combines a chart review with newly recruited patients. This creates a problem as the methods section technically must be reproducible. There is no way of knowing if the same laboratory methods, under similar conditions was used to generate data collected from the chart review. The study can be salvaged by describing either of the groups independently as two studies and not combining their data.

Data Analysis, Discussion and Conclusion: Due to the anomaly in the methods section, the data analysis, discussion and conclusion are not valid.

 The prospective and retrospective participants described separately Page 10-13

Reviewer #2: 

Introduction 

1. The general situation of HBV treatment in Tanzania needs to be clarified. Are Bugando Medical Center and Muhimbili National Hospital the only sites that treat HBV in Tanzania? YES. Currently HBV treatment is available at Bugando Medical centre and Muhimbili National Hospital. The statement re-written to reflect the concept. Page 4, 5

2. Is there any information on the number of individuals on treatment among those diagnosed with HBV in Tanzania?

 Data on HBV treatment is scarce in Tanzania. The MNH project data have not yet published. Negligible information is available from Bugando Medical Centre. Page 4

Methodology 

3. I suggest that you briefly explain the principle of the formula used to calculate the sample size, including the numbers. Responded accordingly Page 9, 10

4. Is there information in the database about testing for other pathogens such as HIV, HCV, and HDV? Is testing routine in Tanzania?

 HIV and HCV are routinely done in Tanzania, however in the prospective patients, these tests were not done. Also in the retrospective patients, these data were not available Page 11, 12

5. Is there data on the algorithm used for HBsAg testing in the patients included in the study? Which test is used?

 There is HBsAg testing algorithm in Tanzania. Tanzania follows the 2015 WHO guidelines for HBsAg testing, treatment and monitoring of HBV management Page 5, 6

Results

6. Tables 1 and 2 could be merged into a single table. This would give a better view of the results.

 Table 1 and 2 combined (now table but with three columns) Page 15, 16,17

7. Tables 1 and 2 can have two columns. One with the characteristic and the other with the absolute number and percentage, just indicate it in the column legend. Table 1 and 2 combined (now table but with three columns) Page 15, 16, 17

8. Other factors such as co-infection with HBV, HDV, and HCV were not taken into account in the eligibility for treatment. If there is no data, it is important to discuss it.

 Coinfection of HBV with HIV, HDV and HCV were not available on routinely testing as explained in the manuscript. These limitations are well discussed Page 11, 12, 26, 27

Discussion 

9. The WHO recommendations for treatment were updated in 2024. The study was carried out under the 2015 recommendations, but it is also important that the results are discussed in the context of the new recommendations. It was considered accordingly Page 4, 6, 7, 17, 19

---

## [Decision Letter · Decision Letter 1]

9 Aug 2024

Hepatitis B infection: Evaluation of demographics and treatment of chronic hepatitis B infection in Northern-western Tanzania

PONE-D-24-17908R1

Dear Dr. Mlewa,

We’re pleased to inform you that your manuscript has been judged scientifically suitable for publication and will be formally accepted for publication once it meets all outstanding technical requirements.

Kind regards,

Jason T. Blackard, PhD

Academic Editor

PLOS ONE

Additional Editor Comments (optional):

None

Reviewers' comments:

Reviewer's Responses to Questions

**Comments to the Author**

1. If the authors have adequately addressed your comments raised in a previous round of review and you feel that this manuscript is now acceptable for publication, you may indicate that here to bypass the “Comments to the Author” section, enter your conflict of interest statement in the “Confidential to Editor” section, and submit your "Accept" recommendation.

Reviewer #1: All comments have been addressed

Reviewer #2: (No Response)

2. Is the manuscript technically sound, and do the data support the conclusions?

Reviewer #1: Yes

Reviewer #2: Partly

3. Has the statistical analysis been performed appropriately and rigorously? 

Reviewer #1: Yes

Reviewer #2: Yes

4. Have the authors made all data underlying the findings in their manuscript fully available?

Reviewer #1: Yes

Reviewer #2: Yes

5. Is the manuscript presented in an intelligible fashion and written in standard English?

Reviewer #1: Yes

Reviewer #2: Yes

6. Review Comments to the Author

Reviewer #1: the reviewers comments have been addressed. The authors show no competing interests. Ethical approvals have been obtained and are in line with the Helsinki declaration.

Reviewer #2: Most of the questions have been answered. However:

Introduction

Figure 1/Table 1 is public, so it is unnecessary to include it in the manuscript.

Methodology/results

The sample explanation was clear, however, the combined analysis of data from retrospectively and prospectively recruited patients makes the analysis and interpretation of the results weak.

7. PLOS authors have the option to publish the peer review history of their article (what does this mean?). If published, this will include your full peer review and any attached files.

Reviewer #1: No

Reviewer #2: No

---

## [Editor Report · Acceptance letter]

15 Aug 2024

PONE-D-24-17908R1 

PLOS ONE

Dear Dr. Mlewa, 

I'm pleased to inform you that your manuscript has been deemed suitable for publication in PLOS ONE. Congratulations! Your manuscript is now being handed over to our production team.

Kind regards, 

on behalf of

Dr. Jason T. Blackard 

Academic Editor

PLOS ONE